# Unexpected Discovery of *Thelypteris palustris* (*Thelypteridaceae*) in Sicily (Italy): Morphological, Ecological Analysis and Habitat Characterization

**DOI:** 10.3390/plants10112448

**Published:** 2021-11-12

**Authors:** Saverio Sciandrello, Salvatore Cambria, Gianpietro Giusso del Galdo, Gianmarco Tavilla, Pietro Minissale

**Affiliations:** Department of Biological, Geological and Environmental Sciences, University of Catania, Via A. Longo 19, I-95125 Catania, Italy; cambria_salvatore@yahoo.it (S.C.); g.giusso@unict.it (G.G.d.G.); gianmarco.tavilla@phd.unict.it (G.T.); p.minissale@unict.it (P.M.)

**Keywords:** distribution, ecology, relic fern, Mediterranean wetlands, conservation status, pteridophytes, vegetation

## Abstract

*Thelypteris palustris* Schott (*Thelypteridaceae*), known as “marsh fern”, is infrequent in the Mediterranean area. The occurrence of this species is known for almost all the Italian regions (except for Sardinia and Sicily), but with rare and declining populations. During floristic fieldwork on the Sicilian wetlands, a new unknown population was found. The aim of this paper is to analyze the morphological traits of the species, as well as its ecological features and the floristic composition of the plant communities where it lives. According to IUCN guidelines, here we provide the regional assessment (Sicily) of *T. palustris*. To analyze its morphological features, many living plants were examined, with particular attention to the spore structure. A total of **179** plots (110 species) and **34** pools were sampled. Our results highlight the relic character of the species which is at the southernmost border of its distribution range. The micro-morphological investigations on the spores show that the Sicilian population belongs to the subsp. *palustris*. The floristic analysis confirms the clear dominance of perennial temperate-cold zones Eurasian taxa. Finally, a new association, *Thelypterido palustris-Caricetum paniculatae*, within the *Caricion gracilis* alliance (*Phragmito-Magnocaricetea* class) is described.

## 1. Introduction

*Thelypteris palustris* Schott (*Thelypteridaceae*), known as “marsh fern”, is a deciduous species that represents one of the most complex species in the pteridophytes. Fernald [1] recognized four varieties of *Thelypteris palustris*: var. *palustris* of Eurasia (from Europe and NW Africa to eastern Himalayas and southern China); var. *pubescens* (G. Lawson) Fernald of northeastern United States, Canada, and eastern Asia; var. *haleana* Fernald of the southeastern United States and Bermuda; and var. *squamigera* (Schltdl.) Weath. of Africa, southern India, northern New Zealand [2]. Afterwards Tryon et al. [3], mainly analyzing the spore structure, recognized two species, one largely of the southern hemisphere (*T. confluens* (Thunb.) C.V.Morton = *T. palustris* var. *squamigera*) and the other, *T. palustris*, including two varieties (*T. palustris* var. *palustris* and *T. p.* var. *pubescens*), in the northern hemisphere.

This taxonomic view has been confirmed in recent times, so currently the genus *Thelypteris* includes two species, *T. palustris* in the northern hemisphere and *T. confluens* in the southern hemisphere, the only change regarding the rank of the two taxa within *T. palustris* that are now considered subspecies [4].

In Europe *Thelypteris palustris* (subsp. *palustris*) is known from several countries [5].

In Italy, this species is known for almost all regions, except for Sardinia and Sicily, although it is reported as an extinct or doubtful taxon for many territories [6,7,8].

The species has undergone a considerable decline throughout its distribution range, mainly due to habitat loss and reduction. Despite this significant decrease in area at the European level, it was recently classified as Least Concern (IUCN category) [9].

In northern Europe *Thelypteris palustris* has been found in several plant communities, for example in open habitats and in clear woodland: *Juncus subnodulosus* Schrank fen-meadow, *Salix cinerea* L. woodland, *Alnus glutinosa* (L.) Gaertn. woodland, *Betula pubescens* Ehrh. woods [10]. In Italy, the species has been found in the plant communities of the *Phragmito-Magnocaricetea* [11,12,13], and in the swamp forests of the *Alnetea glutinosae* class [14,15,16,17,18,19,20,21].

Our finding, during a survey in the Nebrodi Mountains (northern Sicily), is very interesting because the species is quite rare in the Italian territory since the habitats where it grows are in strong reduction, and even more because the Sicilian population represents the southernmost limit of its distribution range. The new finding is certainly unexpected because the flora of Sicily is one of the best studied in Italy and probably in Europe: consider that the start of a “modern” botanical exploration of the island date back to 1664 at least [22]

Marginal habitats in the Mediterranean area represent sites of high ecological importance and a refuge for threatened plants (e.g., hygrophytes) like the case of *Thelypteris palustris*. In fact, these hydrophytic species are linked to peculiar ecological requirements and are highly susceptible to climate changes, and this could be led to their disappearance in the next years. The correct identification of *T. palustris*, as well as the floristic composition of the plant community where it grows and its ecology, are relevant issues for future conservation measures and monitoring actions of this species.

The main objectives of this research are split up into two parts. One is to examine the morphological and ecological features of the new population of *T. palustris*, as well as to assess its conservation status in Sicily. The second one is to provide data about the habitat where *T. palustris* grows and to analyze the floristic composition of the plant community.

## 2. Results and Discussion

### 2.1. Description of the Species (Based on the New Population)

*Thelypteris palustris* Schott, Gen. Fil. [Schott] ad t. 10 (1834) subsp. *palustris* (Figure 1 and Figure 2) Plant terrestrial. **Rhizomes** long creeping, black, glabrous, with more or less solitary leaves. **Fronds** monomorphic, 40–60 cm long; **petioles** 20–36 cm long, bases black, polished, usually glabrous, or rarely with sparse scales, 2.7 × 1.6 mm, irregular to elliptic-lanceolate, light-brown to yellowish, adpressed to patent; **laminae** lanceolate 20–28 cm long, 8–12 cm wide, 1-pinnate-pinnatifid, apices shortly acuminate and pinnatifid; **Rachises** with sparse whitish hairs, 0.1–0.4 mm long; **pinnae** 18–20 pairs, subopposite, flat- or obliquely spreading, usually slightly reflexed, short-petiolulate 0.45–0.50 mm wide, 0.7–0.9 mm long; proximal pair slightly shortened, **middle pinnae** lanceolate, 4.5–5.8 × 0.8–1.4 cm, bases truncate, pinnatifid nearly to costae, apices shortly acuminate; **segments** 4.2–6.8 × 2.1–2.8 mm, rounded-obtuse or obtuse-pointed at apices, fertile segments usually recurved to forming points along margin. **Veins** pinnate in segments, lateral veins 6–8(9) pairs, forked and reaching margins, proximal pair arising from base of costa. **Laminae papery**, grass-green or yellowish green when dry, glabrous on both surfaces, rachises and costae grooved adaxially, raised abaxially, glabrous on both sides or with acicular long hairs abaxially. **Sori** orbicular, dorsifixed at middle of veinlets, located between costa and margins; **indusia** small, orbicular-reniform, membranous, deciduous when mature. **Spores** ca. 45 × 30 μm, with papillate surfaces, papillae 3–5 μm high, perforated at the base. Terrestrial in swamps, bogs, and marshes, also along riverbanks and in wet woods; 0–1400 m.

### 2.2. Distribution and Conservation Status in Italy

In Italy, the species is reported for almost all regions. Probably, it has never been found in some territories due to the reduction or disappearance of its natural habitat, or in some cases, also owing to incorrect reports. It is reported as extinct in Marche, as a doubtful record in Molise and Campania, and not found in recent times in Umbria, Valle d’Aosta and Abruzzo [7,8,9]. In southern Italy the species is highly localized, with an altitude range between 0 and 1000 m a.s.l., from the coast to the mountain, occurring in Puglia at Laghi Alimini, Otranto [23,24], in Calabria at Lago dell’Aquila, Reggio Calabria [13,25] and in Sicily at Serra della Testa (Nebrodi) (Figure 3).

This population recently discovered in Sicily has extended its distribution range and represents the southernmost population of Italy.

According to the European Red List of Vascular Plants [26], the species is classified as Least Concern (**LC**). Currently, in Italy, *Thelypteris palustris* has been recently evaluated as vulnerable (**VU**) by Orsenigo et al. [27] based on the criterion B [28]. In Sicily, the total area occupied by *Thelypteris palustris* is about 0.62 ha. Despite its very small distribution area, it was not possible to carry out a detailed count of the individuals of the population due to the stoloniferous vegetative development of the species. Therefore, thanks to our data and according to the IUCN criterion B, we recommend considering *Thelypteris palustris* as Critically Endangered (**CR B2abii, iii, iv**) for Sicily, due to a very small AOO (4 km^2^), the occurrence on one location, and possible decline of the population especially because of the grazing practices and water flow reduction due to climate change.

### 2.3. Plant Communities with T. palustris in Italy

*Thelypteris palustris* is indicated as a characteristic/diagnostic species of the swamp forests of the *Alnion glutinosae* alliance *(Alnetea glutinosae* class). In Italy, especially in the northern sector, several plant communities of the *Alnetea glutinosae* class include *Thelypteris palustris*, such as *Carici acutiformis-Alnetum glutinosae* Scamoni 1935 [16]; *Carici elatae-Alnetum glutinosae* Franz ex Sburlino, Poldini, Venanzoni et Ghirelli 2011 [17,18,19,20]; *Carici elongatae-Alnetum glutinosae* Tüxen 1931 [29]; *Thelypterido-Alnetum glutinosae* Klika 1940 [14,15]; *Rhamno catharticae-Ulmetum minoris* Poldini, Vidali, Castello, Sburlino [30]; *Hydrocotylo vulgaris-Alnetum glutinosae* Gellini, Pedrotti ex Venanzoni, 1986 [21,31]; *Limnirido pseudacori-Fraxinetum oxycarpae* Gennai, Gabellini, Viciani, Venanzoni, Dell’Olmo, Giunti, Lucchesi, Monacci, Mugnai et Foggi 2021 [21]; *Cladio marisci–Fraxinetum oxycarpae* Piccoli, Gerdol & Ferrari ex Piccoli 1995 [21]; *Valeriano dioicae-Fraxinetum oxycarpae* Poldini et Sburlino 2018 [21].

*Thelypteris palustris* is reported, also, for Lake Massaciuccoli (northern Tuscany), in peculiar reed-beds (*Thelypterido palustris-Phragmitetum australis*) developing on floating islands rich in decaying organic matter [12]. This community was included in the *Carici pseudocyperi-Rumicion hydrolapathi* alliance (*Magnocaricetalia elatae, Phragmito-Magnocaricetea*). Probably, also the community of Lake Alimini (Otranto, Lecce) [23,24], which hosts *Thelypteris palustris*, is to be referred to *Thelypterido palustris-Phragmitetum australis*.

Moreover, the species is reported for Lake Aquila (Calabria), in tall sedges marsh vegetation (*Cladietum marisci* Allorge 1921) included in the *Magnocaricion elatae* alliance (*Magnocaricetalia elatae* Pignatti 1954) [13].

In Sicily, *Thelypteris palustris* falls within the sedges of the *Caricion gracilis* alliance (*Magnocaricion elatae*). This alliance, until now never reported in Sicily, groups plants communities growing on eutrophic clayey soils flooded for long time with a temperate Europe distribution.

### 2.4. Vegetation Ecology and Habitat

Overall, **15** different plant communities, each one with specific floristic compositions, were identified (Appendix B). Most of these plant communities were investigated by Brullo et al. [32] for Nebrodi Mounts. Therefore, we avoid a detailed description of the investigated communities. The wide sampling and cluster analysis allowed us to highlight the uniqueness and rarity of *T. palustris* in Sicily and define objectively the correct syntaxonomic framework. The cluster analysis of all relevés carried out on the Nebrodi Mounts showed 2 main groups (Figure 4). The first group (**cluster A**) includes mainly the helophytic perennial vegetation of the *Phragmito-Magnocaricetea* class, while the second group (cluster B) includes the aquatic vegetation of the *Lemnetea* and *Potametea* classes [31]. Within the *Phragmito-Magnocaricetea* four alliances can be distinguished: the first one (**A11**) *Phragmition communis* includes the vegetation dominated by tall graminoid species subjected to regular, prolonged periods of flooding that grow on mineral meso-eutrophic, often muddy, soils; the second one, *Magnocaricion elatae* (**A121**) consist of plant communities of mesotrophic to dystrophic soils, often peaty and flooded for prolonged periods; the third alliance *Caricion gracilis* (**A122**) groups communities of eutrophic soils, flooded for prolonged periods; the fourth alliance *Alopecuro-Glycerion spicatae* (**A2**), that includes the vegetation of hygrophilous herblands of shallow montane pools characterized by large water-depth fluctuations at high altitudes of Sicily. This last alliance is grouped with a peculiar annual amphibious vegetation dominated by *Lythrum portula* which falls within the *Nanocyperetalia* order (*Isoeto-Nanojuncetea*). Within the second group (**cluster B**) two subclusters can be distinguished: the first one (**B1**) (*Potametea pectinati*) delimits the perennial macrophytic communities of fresh, mesotrophic to eutrophic, waters; while the second one subcluster (**B2**) includes (*Lemnetea minoris*) the floating pleustophyte communities eutrophic to hypertrophic waters.

Bray-Curtis ordination shows a marked correspondence with cluster analysis (Figure 5). The highest data dispersion is obtained with axes 1 and 2. On the positive side of axis 1 there are the helophytic perennial vegetation of the *Phragmito-Magnocaricetea* class with high floristic diversity values, while on the negative side of axis 1 are distributed the aquatic vegetation of the *Lemnetea* and *Potametea* classes, with low values of floristic diversity. The *Carex paniculata* community (cluster 7) is very isolated from the other associations, probably due to the peculiar ecological and floristic conditions of the wet habitat.

### 2.5. Floristic Composition and Phytosociological Insights of the Thelypteris palustris Population in Sicily

In the study area, *Thelypteris palustris* was found exclusively in a perennial wetland characterized by *Carex paniculata* L. and *Juncus subnodulosus*. This perennial vegetation grows on flat or slightly sloping surfaces, on clayey-silty acid soils, permanently wet and rich in organic matter. The structure is determined mainly by *Carex paniculata*, the dominant species in terms of biomass and number of individuals, joined to several hygrophilous species, as *Galium palustre* L. subsp. *elongatum*, *Mentha aquatica* L., *Cirsium creticum* (Lam.) d’Urv. subsp. *triumfettii* (Lacaita) K.Werner, *Juncus subnodulosus*, *Carex distans* L., *Cyperus longus* L., *Hypericum tetrapterum* Fr., *Phragmites australis* (Cav.) Trin. ex Steud., *Epilobium parviflorum* Schreb., *Lolium arundinaceum* (Schreb.) Darbysh., *Eupatorium cannabinum* L., *Lotus rectus* L., *Eleocharis palustris* (L.) Roem. & Schult., *Helosciadium nodiflorum* (L.) W.D.J. Koch, *Rumex conglomeratus* Murray. The constant presence of *Thelypteris palustris* highlights the mesophilous character of the plant community, clearly differentiating it from the other sedge communities present in central and northern Italy. Therefore, because of its ecological features, *Thelypteris palustris* is proposed as a characteristic species of a new association named *Thelypterido palustris-Caricetum paniculatae* ass. nova hoc loco (Table 1, Rel. 10, cluster 7) included in the *Caricion gracilis* alliance and *Magnocaricetalia elatae* order (*Phragmito-Magnocaricetea* class. The new association is also characterized by a floristic component of the *Holoschoenetalia vulgaris* order, as *Lysimachia nemorum* L., *Holcus lanatus* L., *Dactylorhiza maculata* (L.) Soó subsp. *saccifera* (Brongn.) Diklić, *Juncus effusus* L., *Lythrum junceum* Banks & Sol. This later order includes hygrophilous communities dominated by helophytes (rushes and sedges) that grow in depressions in the supratemperate thermotype subjected to periodic submersions, on soils with low permeability and a rich silty-clayey component [33,34,35]. In addition, the association hosts floristic elements, very rare in Sicily, of high phytogeographic value (Figure 6), such as *Epipactis palustris* (L.) Crantz, *Equisetum palustre* L., *Rhynchocorys elephas* (L.) Griseb., *Juncus conglomeratus* L., *Carex flacca* Schreb. subsp. *flacca, C. pallescens* L., etc. The muscinal component also plays an important ecological role, particularly *Calliergonella cuspidata* (Hedw.) Loeske with a high degree of coverage and sociability. From the chorological and structural viewpoint, this vegetation highlights the relevance of the species with an Euroasiatic-Circumboreal distribution (34%), with geophytes (34%) and hemicryptophytes (61%) being the dominant life forms. This new association can be considered a southern vicariant of the *Caricetum paniculatae*, with a central and northern Italian distribution [36,37]. This last association shows structural affinities with *Thelypterido palustris-Caricetum paniculatae* owing to a high cover of *Carex paniculata*. However, the two plant communities can be clearly separated, based on many differential diagnostic species, such as *Epipactis palustris*, *Equisetum palustre, Juncus subnodulosus*, and *Rhynchocorys elephas*. From a bioclimatic point view the *Thelypterido palustris-Caricetum paniculatae* falls into the lower Supramediterranean belts with lower subhumid ombrotype [38], in contact with deciduous thermophilic *Quercus cerris* oak forests, referable to the *Arrhenathero nebrodensis-Quercetum cerridis* [39].

## 3. Materials and Methods

### 3.1. Study Area

The study area is situated in the Nebrodi Mounts, Sicily’s largest mountain complex (Figure 7). They are located in the N-E part of the island, between the west side of the Peloritani Mountains and the east side of the Madonie Mounts, constituting the extension of the Apennine ridge on the island. They are a mountain range without major roughness that reaches its maximum altitude at Monte Soro (1847 m a.s.l.). From a geological point of view, this territory is mainly made up of sedimentary successions belonging to different periods. The dominance of Flysch is mostly noted, the oldest sediments belonging to the Alpine Tethys Units [40], they are Cretaceous in age and are represented by deep-water flyschs and scaly clays. Most of the outcropping rocks are part of the so-called Flysch of Monte Soro (upper Tithonian, lower Cretaceous) and Numidian (lower Oligocene Miocene) [41].

The outcrop of clayey layers favors the formation of humid environments and ponds, and lakes originate where the orographic conditions allow it. The existence of humid environments on the Nebrodi is possible because of favourable climatic conditions that characterize this mountain area that is the most mesic and rainy in Sicily, being affected by average annual rainfall between 1000 and 1400 mm. According to Rivas Martínez et al. [42], the bioclimate of this area is supra-Mediterranean lower middle-humid bioclimatic conditions [38]. Although these small wetlands can have a relatively short lifespan due to landfills, climate changes, etc., for some of these areas in the Nebrodi Mounts an existence has been documented since the end of the last glaciation (about 10,000 years ago) when it seems that the climate had become wetter in Sicily [43]. The climatic and geomorphological conditions of the Nebrodi Mounts make it the area with the greatest wooded coverage and with the highest values of biodiversity in Sicily [44]. In particular, this territory is characterized by very extensive oak forests (*Quercus cerris* L.) at medium altitudes, and beech woods (*Fagus sylvatica* L.) at higher altitudes. However, grazing meadows and small wetlands (mostly natural) are the main discontinuities in the forest cover of this territory.

### 3.2. Data Sets and Data Processing

The morphological study regarding *Thelypteris palustris* was carried out on living material (15 specimens), all coming from Nebrodi Mounts territory. The collected samples were kept at the Catania Herbarium (CAT). For scanning electron microscopy (SEM) images, samples of spores were transferred from herbarium specimens to aluminum SEM stubs coated with double sided carbon tape. The stubs were then sputter-coated with gold and imaged digitally using a Zeiss EVO LS10, with an accelerating voltage of 30 kV, in the Center for Microscopy at the University of Catania. The morphological terminology used in the description follows Lellinger [45], while to spore nomenclature follows Tryon & Lugardon [46].

To analyze the structure and floristic composition of the marsh vegetation in the Nebrodi Mounts, **34** pools were examined. A total of **179** unpublished phytosociological relevés (**110** species) were collected, personally sampled in the period April 2018-June 2021. The floristic composition and cover of species in each plot were determined by using the standard method of relevés [47]. All the relevés were classified using classification and ordination methods. Numerical analysis was performed using the software package “PC-ORD”, 6.08 software. A multivariate analysis (Linkage method: Ward’s, Distance measure: Sorensen (Bray-Curtis) was applied. Bray-Curtis ordination (Distance measure: Jaccard) takes into account different quantitative data, such as vegetation coverage (%), altitude, number of species (N. sp.), Altitude (m a.s.l.), Slope (°), Aspect, and Simpson/Shannon index. Quantum GIS software version 3.6 and GPS Garmin Montana was used to geolocate the surveyed wetlands.

For the risk assessment at the regional scale (Sicily), we followed the IUCN protocol and the most recent guidelines for its application [28]. In particular, we applied the IUCN criterion B by estimating trends in the Area of Occupancy (AOO), that is, the area covered by a taxon. AOO was assessed by using a 2 × 2 km grid [48]. Syntaxa classification follows Biondi et al. [33], and Mucina et al. [49]. Taxonomic nomenclature follows Bartolucci et al. [9] and Pignatti [50,51,52,53].

## 4. Conclusions

The species we found in Sicily seems to be very rare in the island, unless new discoveries that might be made in the future. It is localized in a microrefuge area that means, according to Rull [54], a small area with local favorable environmental features, in which small populations can survive outside their main distribution area, protected from the unfavorable regional environmental conditions. *T. palustris* grows in contexts which, even if somewhat subject to disturbing factors such as grazing, maintain good natural characteristics. However, these places are vulnerable to further disturbances, such as drainage, and above all to the decrease in rainfall triggered by climate change as detected for Sicily [55,56] that could jeopardize its precarious survival. Therefore, even the microrefuge area may not be enough to guarantee the existence in Sicily of this species; in any case, it will need to be monitored over time. This is a general trend that can undermine a species that, although with a large distribution range, is linked to peculiar environmental conditions. These circumstances could fail especially in semi-arid areas such as around the Mediterranean basin where climate change overlaps the usual intense anthropogenic disturbance that particularly affects wetlands [57]. In the Mediterranean area, the populations of *T. palustris* are likely to be declining following the general trend of destruction and degradation of shallow wetlands. It is not considered common anywhere in its Mediterranean range. It is very rare in Morocco and in Algeria. In Morocco it is known from two localities only: Bou Charen and near to Açilah in the western Rif. In Algeria, *T. palustris* is known from three localities, including Senhadja in Numidie. It is widespread in Turkey, but its habitats are under threat and because of this in the future this taxon may be threatened [58]. In Italy, as we have shown, many reports are old and no longer reconfirmed. The species, going south, is highly localized, occurring only in Puglia, Calabria, and Sicily (Figure 3), with only one sub-population for each region. Therefore, this recently discovered in Sicily extend its distribution area and represents the southernmost population of Italy.

Plants characterizing wet environments represent one of the most threatened groups of the Mediterranean flora [59,60,61,62,63,64,65]. For the reasons quoted above, these areas require urgent and effective conservation policies not only to safeguard the biodiversity but also for the important ecosystem services they perform [66]. An example can be our study that illustrates the environmental context of the *T. palustris* populations and the morphological features of the species. Moreover, it clarifies some ecological requirements which are relevant issues for future conservation measures for this species, especially in the Mediterranean areas where the extremely scattered distribution with isolated populations make it vulnerable to disappearance. Unlike the northern European populations, quite widespread, the risk of local-scale extinction is really high in all the Mediterranean populations.

The floristic composition of the Sicilian *Thelypteris palustris* plant community shows a clear affinity with the common sedge communities of the northern Europe. Perhaps, this plant community is a relict vegetation type of the last glacial stage, which currently is localized exclusively in the humid stands of the Nebrodi Mounts. These microrefuges were originated by peculiar geological characteristics of the territory, with a humid supramediterranean bioclimate that facilitate the growth of these hygrophilous species [43].

In conclusion, our study has made possible to highlight the unexpected occurrence in Sicily of the marsh fern *Thelypteris palustris*, growing together with some floristic elements of the highest nature value, such as *Equisetum palustre, Epipactis palustris, Utricularia australis* R. Br., *Rhynchocorys elephas*, *Juncus conglomeratus*, *J. subnodulosus*, *Carex paniculata*, *C. flacca* subsp. *flacca, C. pallescens*, and *Hypericum tetrapterum*. These vascular species, linked to wetlands, show in Sicily a narrow distribution range due to a strong reduction of their habitat in recent decades. Although they are included in the “A” zone of the Nebrodi Park (Figure 7) and within the Natura2000 site SAC ITA030014, targeted conservation and monitoring actions would be desirable, aimed at the long-term conservation of the floristic component and especially their humid habitats.

## Figures and Tables

**Figure 1 plants-10-02448-f001:**
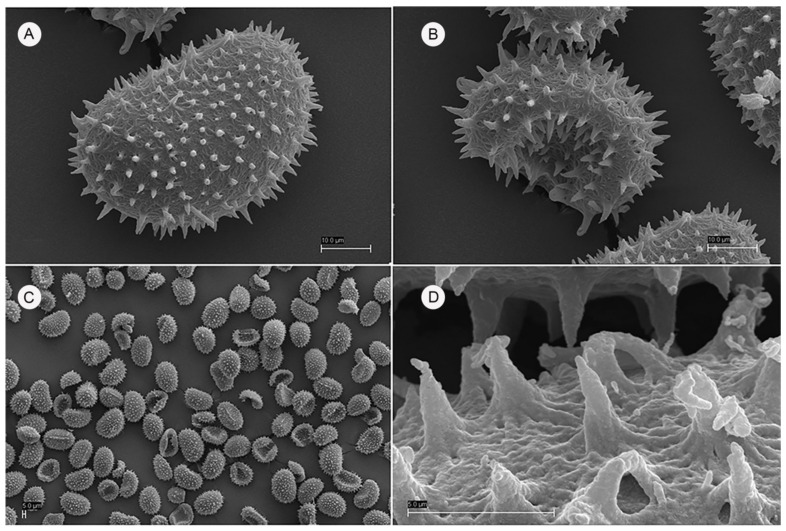
*Thelypteris palustris* Schott subsp. *palustris* spores from the new Sicilian population: (**A**) Spores with diffuse echinate elements; (**B**) Lower spore structure; (**C**) Echinate spores in group; (**D**) Echinate sculpture.

**Figure 2 plants-10-02448-f002:**
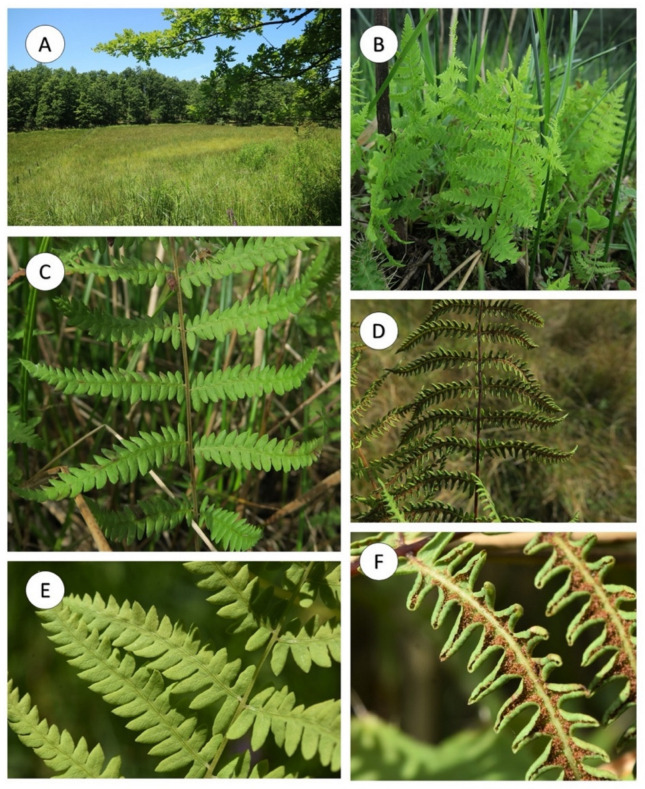
Some views of the new Sicilian population of *Thelypteris palustris* Schott subsp. *palustris*; (**A**) Growth environment (Nebrodi Mountains); (**B**–**E**) Habit; (**F**) Sporangia (Photos of the Authors).

**Figure 3 plants-10-02448-f003:**
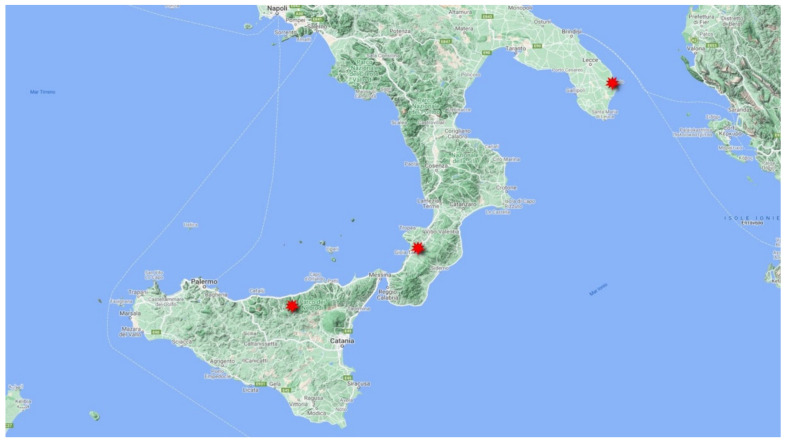
Current distribution of *Thelypteris palustris* in southern Italy.

**Figure 4 plants-10-02448-f004:**
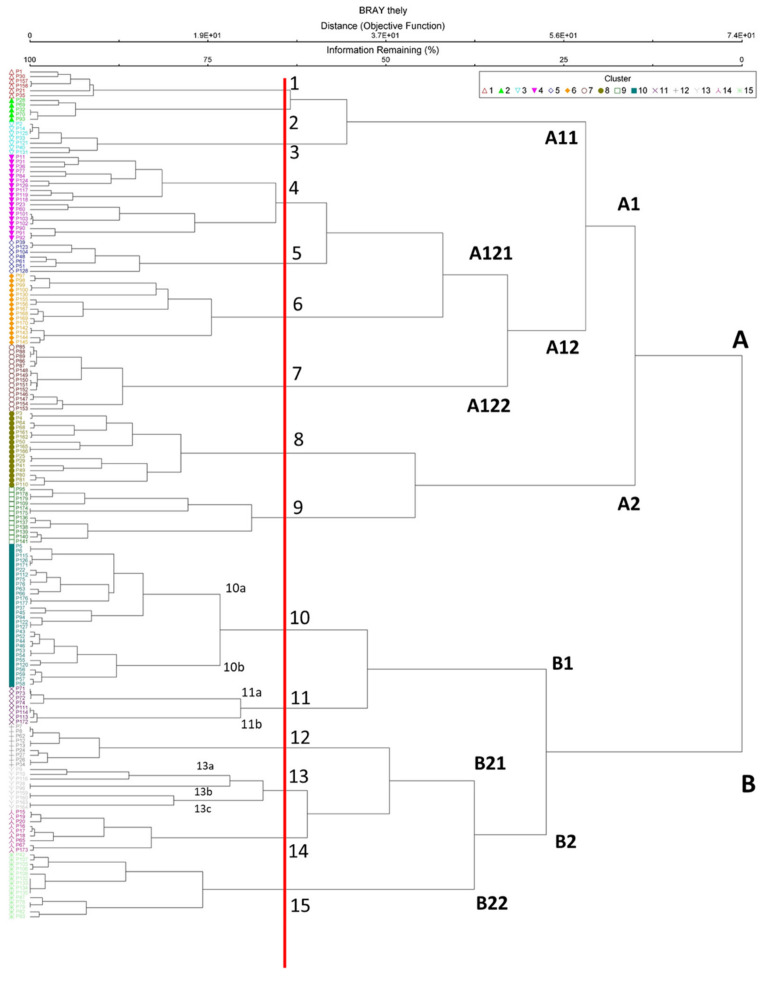
Cluster analysis of 179 unpublished phytosociological relevés. Plant communities: 1. *Sparganietum erecti*; 2. *Scirpetum lacustris*; 3.* Typhetum domingensis*; 4. *Galio palustris-Juncetum inflexi*; 5. *Eleocharitetum palustris*; 6. *Iridetum pseudacori*; 7. *Thelypterido palustris-Caricetum paniculatae*; 8. *Lythrum portula* comm.; 9. *Glycerio spicatae-Oenanthetum aquaticae*; 10a. *Potametum natantis*; 10b. *Utricularietum australis*; 11a. *Potamogetono natantis-Polygonetum natantis*; 11b. *Potametum pusilli*; 12. *Myriophylletum verticillati*; 13a. *Lemnetum minoris*; 13b. *Potamogetono-Ceratophylletum submersi*; 13c. *Ranunculetum omiophylli*; 14. *Ranunculetum aquatilis*; 15. *Wolffietum arrhizae*.

**Figure 5 plants-10-02448-f005:**
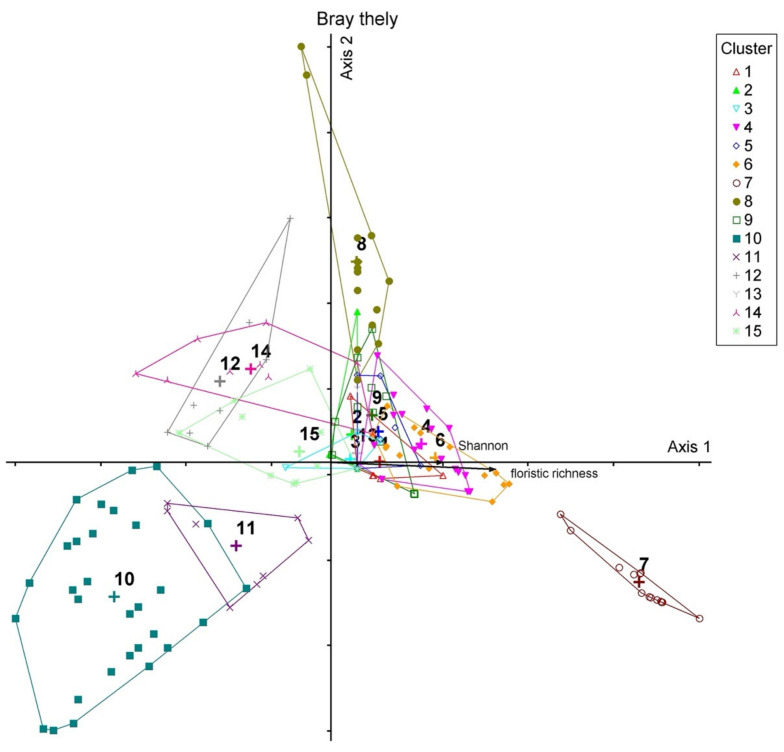
Bray-Curtis ordination. Axis 1 extracted 9.97% of the original distance matrix Cumulative: 9.97%; Axis 2 extracted 4.89% of the original distance matrix. Cumulative: 14.86%. Plant communities according to Figure 3.

**Figure 6 plants-10-02448-f006:**
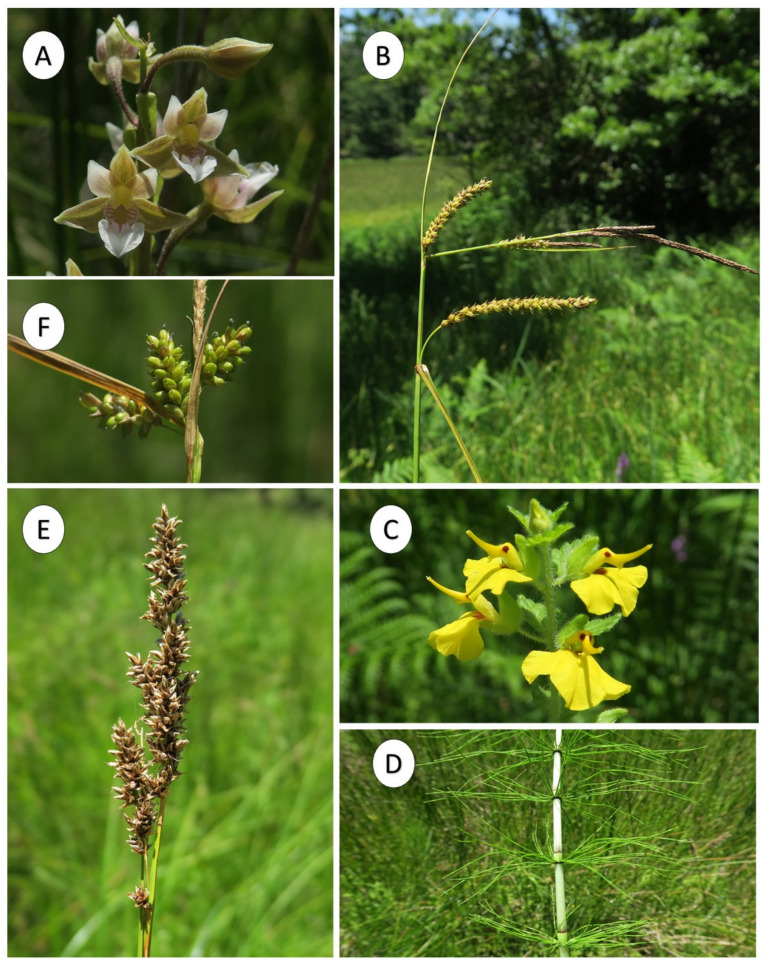
**Photo plate illustration of some rare hygrophilous species of the Nebrodi Mountains:** (**A**) *Epipactis palustris*; (**B**) *Carex flacca* subsp. *flacca*; (**C**) *Rhynchocorys elephas*; (**D**) *Equisetum palustre*; (**E**) *Carex paniculata*; (**F**) *Carex pallescens*. (Photos of the Authors).

**Figure 7 plants-10-02448-f007:**
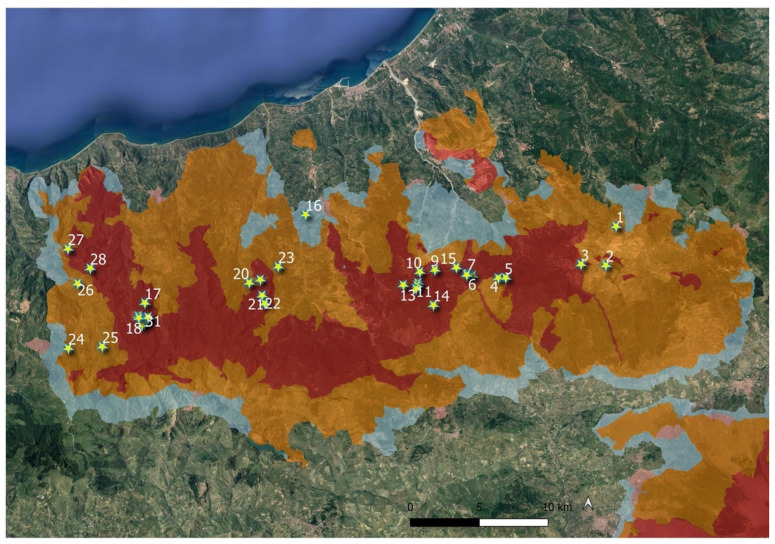
Wetlands investigated in Nebrodi Mounts (localities are given in Appendix A), with zonation of the regional park (zone A: red; zone B: orange; zone C: blue; zone A is the most protected zone).

**Table 1 plants-10-02448-t001:** *Thelypterido palustris-Caricetum paniculatae* ass. nova hoc loco–Sampling plots main features of plant community investigated.

		Pools	18	18	18	18	18	18	18	18	18	18	18	18	18	31	
		Relevé number	**1**	**2**	**3**	**4**	**5**	**6**	**7**	**8**	**9**	**10**	**11**	**12**	**13**	**14**	
		Altitude (m a.s.l.)	1000	1000	1000	1000	1000	1030	1030	1028	1028	1028	1000	1000	1028	1057	
		Surface (mq)	50	50	50	50	50	50	50	50	50	50	50	50	50	50	
		Coverage(%)	100	100	100	100	100	100	100	100	100	100	100	100	100	100	
		Slope (°)	-	-	-	-	-	-	-	5	5	5	5	5	5	-	
		Aspect	-	-	-	-	-	-	-	W	W	W	W	W	W	-	
		Vegetation height (m)	1.5	1.3	1.3	14	1.5	1.5	1.3	1.3	1.4	1.4	1.4	1.3	1.2	1.5	
		No. species	18	19	20	19	20	25	27	23	24	27	28	24	25	23	
		Simpson_1-D	0.9	0.9	0.9	0.9	0.9	0.95	0.95	0.95	0.95	0.95	0.96	0.95	0.95	0.95	
		Shannon_H	2.8	2.8	2.9	2.9	2.9	3.1	3.2	3.1	3.1	3.2	3.2	3.1	3.1	3.1	
		Evenness_e^H/S	0.9	0.9	0.9	0.9	0.9	0.9	0.9	0.9	0.9	0.89	0.9	0.9	0.9	0.9	
		Equitability_J	0.96	0.96	0.97	0.97	0.97	0.97	0.97	0.97	0.97	0.96	0.97	0.96	0.97	0.97	**presence**
**LF**	**Corology**	**Characteristic species**															
G	Subcosmop	*Thelypteris palustris* Schott subsp. *palustris*	2	1	2	1	2	+	+	2	2	3	3	2	1	-	13
G	Circumbor.	*Epipactis palustris* (L.) Crantz	+	+	-	+	-	-	-.	+	-	+	+	-	-	-	
		** *Char. Magnocaricion elatae and Magnocaricetalia* **															
H	Europ.-Caucas.	*Carex paniculata* L.	4	3	2	3	3	4	3	3	4	4	4	3	1	4	14
H	Euri-Medit.	*Galium palustre* L. subsp. *elongatum* (C. Presl) Arcang.	1	1	2	1	1	1	2	2	1	2	1	2	1	1	14
G	Paleotemp.	*Cyperus longus* L.	1	2	1	1	+	-	-	1	1	1	+	1	+	+	12
G	Europ.	*Carex flacca* Schreb. subsp. *flacca*	-	-	+	+	-	-	+	+	-	+	+	-	-	-	6
H	Circumbor.	*Carex pallescens* L.	.	.	+	.	.	+	.	.	.	+	.	.	.	-	3
H	Eurasiat.	*Rumex conglomeratus Murray*	.	.	.	.	.	.	.	.	.	.	.	.	.	+	1
		** *Char. Phragmito-Magnocaricetea* **															
H	Paleotemp	*Mentha aquatica* L.	1	+	2	1	1	1	2	1	1	+	1	1	1	1	14
H	Orof. NE-Medit.	*Cirsium creticum* (Lam.) d’Urv.subsp. *triumfettii* (Lacaita) K. Werner	+	1	+	1	1	2	1	+	1	+	+	1	+	+	14
G	Europ.-Caucas.	*Juncus subnodulosus* Schrank	2	3	3	2	2	1	2	2	1	1	1	3	1	.	13
H	Eurosiber.	*Angelica sylvestris* L.	+	+	+	+	1	1	+	+	1	+	1	1	+	.	13
G	Circumbor.	*Equisetum palustre* L.	1	2	+	1	+	2	1	1	1	1	+	+	3	.	13
H	Euri-Medit.	*Carex distans* L.	2	1	1	1	+	2	1	1	+	1	1	1	.	2	13
H	Paleotemp.	*Hypericum tetrapterum* Fr.	+	+	1	1	1	+	+	.	.	1	1	+	.	1	11
G	Subcosmop.	*Phragmites australis* (Cav.) Trin. ex Steud.	.	+	+	.	+	+	+	+	+	+	+	.	.	.	9
H	Paleotemp.	*Epilobium parviflorum* Schreb.	+	1	+	+	+	.	.	.	.	+	+	+	.	.	8
H	Paleotemp.	*Lolium arundinaceum* (Schreb.) Darbysh.	.	.	.	.	.	+	1	+	+	+	.	+	.	+	7
H	Paleotemp.	*Eupatorium cannabinum* L.	1	1	1	+	1	.	.	.	.	.	+	+	.	.	7
Ch	Medit.	*Lotus rectus* L.	.	.	+	.	.	.	.	.	+	+	1	+	.	.	5
G	Subcosmop.	*Eleocharis palustris* (L.) Roem. & Schult.	.	.	.	.	.	.	.	+	+	+	+	.	.	1	5
H	Euri-Medit.	*Helosciadium nodiflorum* (L.) W.D.J. Koch	.	.	.	.	.	+	+	.	.	.	.	.	1	1	4
G	Subcosmop	*Glyceria spicata* Guss.	.	.	.	.	.	+	+	.	.	.	.	.	+	1	4
		** *Trasgr. Holoschoenetalia vulgaris* **															
H	Europ.-Caucas. Subatl.	*Lysimachia nemorum* L.	2	+	1	1	1	1	1	1	+	1	1	2	2	.	13
H	Circumbor.	*Holcus lanatus* L.	1	+	1	+	+	+	+	+	+	+	+	+	+	.	13
G	Medit.	*Dactylorhiza maculata* subsp. *saccifera* (Brongn.) Diklić	+	+	.	.	.	+	1	+	+	+	1	+	+	1	11
G	*Cosmop.*	*Juncus effusus* L.	.	.	.	+	+	1	2	+	+	.	+	1	1	1	10
G	Eurosiber.	*Juncus conglomeratus* L.	+	.	+	+	+	+	+	.	+	+	+	.	.	.	9
H	NE-Medit.	*Rhynchocorys elephas* (L.) Griseb.	.	.	.	.	.	+	+	1	1	+	1	+	1	.	8
H	Medit.	*Lythrum junceum* Banks & Sol.	.	.	.	.	.	*+*	*+*	+	+	+	+	+	+	1	9
		** *Other species* **															
		*Calliergonella cuspidata* (Hedw.) Loeske	1	1	1	1	1	1	1	1	1	1	1	1	1	1	14
H	Paleotemp.	*Lathyrus pratensis* L.	.	.	.	.	.	.	+	1	2	+	+	+	.	+	7
H/T	Euri-Medit.-Sett.	*Myosotis sicula* Guss.	.	.	.	.	.	+	+	.	.	+	+	+	.	1	6
H	Circumbor.	*Prunella vulgaris* L.	.	.	.	.	.	.	+	+	+	.	+	.	+	.	5
G	Circumbor.	*Juncus articulatus* L.	.	.	.	+	+	.	.	.	.	.	.	.	2	2	4
H	Subcosmop.	*Samolus valerandi* L.	.	.	.	.	.	+	+	.	.	.	.	.	1	+	4
H	Subcosmop.	*Isolepis cernua* (Vahl) Roem. & Schult.	.	.	.	.	.	+	+	.	.	.	.	.	1	+	4
H	Europ.-Caucas.	*Carex remota* L.	.	.	.	.	.	.	.	.	.	.	.	.	3	.	1
H	Paleotemp.	*Trifolium repens* L.	.	+	+	.	.	.	.	.	.	.	.	.	.	.	2
T	Euri-Medit.	*Ranunculus ophioglossifolius* Vill.	.	.	.	.	.	.	.	.	.	.	.	.	+	+	2
G	NE-Medit	*Geranium versicolor* L.	.	.	.	.	.	.	.	.	.	.	.	.	+	.	1
H	Eurasiat.	*Ajuga reptans* L.	.	.	.	.	.	.	.	.	.	.	.	.	.	+	1

**Localities and dates of relevés.** Rel. 1–5, Serra della Testa 3 (inf.); Rel. 6–7, Serra della Testa 3 (sup.) 15.06.2021; Rel. 8–12, Serra della Testa 3 (inf.) 15.06.2021; Rel. 13, Serra della Testa 3 (rigagnolo, tra inf. e sup.) 15.06.2021; Rel. 14, Serra della testa 2 15 June 2021.

## Data Availability

Data is contained within the article or Appendix A.

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
