# Peer review of "Unexpected Discovery of Thelypteris palustris (Thelypteridaceae) in Sicily (Italy): Morphological, Ecological Analysis and Habitat Characterization"

_plants, 2021, doi:10.3390/plants10112448_

Round 1
Reviewer 1 Report
The manuscript is well written, but I think that the right order of paragraphs to understand the text and have a logical sequence is to put "materials and methods" before "results and conclusions".
check and insert the authors of the mentioned species mentioned for the first time in the text
Author Response
All suggestions were accepted. The sequence of material and methods are required by the journal.
Best regards.
Reviewer 2 Report
Unfortunately, this manuscript cannot be accepted for publication in Plants in this form. My comments are presented below.
First of all, the concept of the research is not correct, because authors consider Thelypteris palustris as "globally rare". However, this species is common or even abundant in Ukraine (e.g. https://dx.doi.org/10.14748/ssp.v4i1.3011), Ireland (https://www.npws.ie/sites/default/files/publications/pdf/RL10%20VascularPlants.pdf), and in general in the whole Europe (https://portals.iucn.org/library/sites/library/files/documents/RL-4-022.pdf) or through the whole world (https://www.jstor.org/stable/23310934?seq=1#metadata_info_tab_contents). Without doubts, a certain species could be locally rare or threatened, although in Italy Thelypteris palustris is indeed threatened (VU): https://doi.org/10.1080/11263504.2020.1739165. Nevertheless, authors cannot state that this species is globally rare. Obviously, authors indicate almost all of my suggestions above, but for uncertain reasons, they state on considerable decline throughout its distribution range.
In addition, the scale of this study is very local. Without doubts, these results deserve to be published, but in a journal of national or regional level, but not in international journal, like Plants.
Finally, some figures look to be superfluous, such as figures 1, 2, 5, because they don't present any significant information, which would be necessary for presenting the results of this research. In addition, in section 3.1, I would recommend pay attention to the description of vegetation cover instead of description of geological properties of the study area. There are problems with names of captions of figures/tables. For instance, what does mean the name of Table 1 "Thelypterido palustris-Caricetum paniculatae"? In this form, it is unknown what did authors mean here.
On the basis of my review, I recommend to reject the manuscript from this journal with re-submitting it to the regional/national journal.
Author Response
The suggestions were partially accepted.
Best regards.
Reviewer 3 Report
The manuscript report a marsh fern, newly dicoveried in Sicily.
The authors also analyzed for morphological, ecological and habitat characterization.
However, this manuscript appears to be in the form of 'communication' rather than 'article'.
It is interesting, although the manuscript is poor at writing skill. I think this manuscript is acceptable. It will serve the purpose of the journal of Plants.
"Amazing" This is too much of an expression.
Author Response
All suggestions were accepted.
Best regards.
Round 2
Reviewer 2 Report
Unfortunately, the revised manuscript was slightly improved according to my previous comments. Actually, just minor (cosmetic) corrections were made by authors. So, the manuscript is still of regional (sub-national) scale. Of course, these results are of certain importance connected with a record of this locally rare plant. However, this submission is not suitable for publication in an international journal, such as Plants, where only high-quality results with a good analysis, discussion and interest to international audience deserve to be published. Meanwhile, for results of such character, there are many national journals in Italy, where these results can be published to be found by the task readers, i.e. national botanist teams.